# Molecular Studies of *TCF4* Gene and Correlation with Late-Onset Fuchs Endothelial Corneal Dystrophy in the Greek Population: A Novel Cost-Effective Diagnostic Algorithm

**DOI:** 10.3390/ijms262311356

**Published:** 2025-11-24

**Authors:** Natalia Petri, Angeliki Margoni, Konstantinos Droutsas, Andriana Diamantopoulou, Nikolaos Kappos, Athanasios G. Papavassiliou, Marilita M. Moschos, Christos Kroupis

**Affiliations:** 1Department of Clinical Biochemistry, ‘Attikon’ University General Hospital, Medical School, National and Kapodistrian University of Athens, Chaidari, 12462 Athens, Greece; 2Department of Biological Chemistry, Medical School, National and Kapodistrian University of Athens, 11527 Athens, Greece; 3First Department of Ophthalmology, ‘G. Gennimatas’ General Hospital, Medical School, National and Kapodistrian University of Athens, 11527 Athens, Greece

**Keywords:** FECD, *TCF4* gene, trinucleotide repeat expansion, SNP, genetic profile, molecular diagnosis, prognostic markers

## Abstract

Late-onset Fuchs endothelial corneal dystrophy (FECD) is a hereditary, progressive, bilateral and irreversible disorder that is characterized by thickening of Descemet’s membrane, microscopic collagenous protuberances known as guttae, and accelerated loss of corneal endothelial cells. Patients initially complain of blurred vision, and as the disease progresses, painful epithelial edema develops. Untreated cases of FECD often result in blindness, and then, the only treatment is corneal transplantation. DNA polymorphisms in many genes have been implicated, among them *TCF4* on chromosome 18q, encoding a transcription factor protein E2-2, which is involved in regulating cellular growth and differentiation in the cornea. In our previous published study, we confirmed the association of an intronic *TCF4* SNP (rs613872) with the disease in our population. The purpose of this present study is to further investigate another intronic point of interest in the same gene, the CTG18.1 trinucleotide repeat expansion. DNA was isolated from EDTA blood from a well-ascertained group of 36 Greek patients with FECD (Krachmer scale ≥ 2) and 58 healthy individuals, age- and sex-matched after obtaining their informed consent. STR-PCR and triplet-repeat primed PCR (TP-PCR) were performed, followed by gel electrophoresis and fragment analysis on an ABI SeqStudio genetic analyzer. Our real-time qPCR genotyping method was used for the SNP in the LightCycler (Roche). Statistical analysis of both genetic results was performed with SPSS and SNPStats.

## 1. Introduction

Fuchs endothelial corneal dystrophy (FECD) is a bilateral disease of the innermost layer of the cornea that accounts at 3–11% of elderly population and ranks first in the causes of corneal transplantation worldwide [1,2,3]. FECD is a genetically complex, female predominant disorder that gradually leads to decrease in visual acuity and blindness in later stages [3,4]. Its cytological picture is characterized by the progressive loss of corneal endothelial cells and the formation of guttae that begins in the center of the cornea and radiates towards the periphery. Endothelial loss triggers proliferation and migration of cells that leads to abnormalities in size (polymegalism) and number (pleomorphism), which can be visualized by specular microscopy as a thickening of Descemet’s membrane (DM) and guttate excrescences on DM. Gradual failure of the corneal endothelial pump and secondary corneal edema affects vision quality, whilst guttae induces visual disturbances and provokes contrast sensitivity [1,5,6]. Therapeutic strategies vary depending on the stage and the progression of the disease. Conservative management with topical medications is effective in the early stages, whereas decreased visual acuity and endothelial decompensation suggest keratoplasty. Corneal transplantation is considered the treatment of choice in severe cases, and the Descemet Stripping Automated Keratoplasty (DSAEK) surgical method is successful most of the time [7]. Nevertheless, the risk for transplant failure and emerging precision medicine highlight the need to advocate for a less-invasive and effective targeted therapy based on genetic signature [6,7,8,9,10].

In the past decades, molecular diagnostics have provided novel insights into the pathogenic mechanisms that are involved in FECD. Unraveling the genetic and molecular nature of FECD is a key to an early and accurate diagnosis and reliable prognosis, whilst it promises evolution in novel therapies such as cell therapies [8,9]. FECDs’ clinical heterogeneity is induced by genetic and non-heritable factors such as smoking and UV. The genetic basis of FECD demonstrates a variability in expressivity and incomplete penetrance. Although inheritance models follow an autosomal dominant mode, a significant inter-population variability in the strength of correlations among gene mutations and FECD phenotypes was observed [11]. Thus, FECD is stratified in two globally recognized forms, the early- and the late-onset FECD, which yield different genetic profiles. The rare early-onset form, which affects individuals from an early age, is usually more severe and positively related to mutations in the collagen gene 8A2 (*COL8A2*) on chromosome 1 [2,8,11]. *COL8A2* gene encodes the a2 chain of short chain collagen VIII, which is an extracellular matrix protein and major component of DM. Initial genetic studies revealed that the most common, late-onset FECD is associated with rare mutations in genes such as solute carrier family 4 sodium borate transporter member 11 (*SLC4A11*), transcription factor 8 gene (*TCFG8*), transcription factor 4 gene (*TCF4*), lipoxygenase homology domains 1 (*LOXHD1*) and ATP/GTP binding protein like 1 (*AGBL1*) [1,12,13,14,15]. Although several genes have been correlated with late-onset FECD, which manifests in the late forties, the most prevalent alteration worldwide is a triplet-repeat expansion in the *TCF4* gene [16,17,18,19,20]. *TCF4* gene, or E2-2, is found on chromosome 18 and encodes the E2-2 protein, which belongs to a family of basic helix-loop-helix transcription factors that regulate cellular growth and differentiation. *TCF4* gene is involved in many developmental functions such as programmed cell death, epithelial–mesenchymal transition (EMT) and transforming growth factor-β (TGF-β) signaling pathways [21]. Additionally, an independent and significant correlation was revealed among FECD and four *TCF4* single nucleotide polymorphisms (SNPs) (rs17595731, rs613872, rs9954153 and rs2286812). There is a diversity on the impact of those four SNPs among different ethnic groups. However, rs613872 in the third intron of the *TCF4* gene prevails as the most significant polymorphism, most probably affecting the binding site of important transcription factors [8,11,12,22,23]. Homozygotes in this *TCF4* variant are at a 30-fold increased risk of having FECD compared to the control group [11,12]. Notably, a non-coding CTG trinucleotide repeat in the second intron of *TCF4* has been identified as the main pathogenic variant. It was named CTG18.1 due its location on 18q21.1 and most probably affects *TCF4* expression levels either through different initiation sites or from splicing deregulation or RNA toxicity and/or RAN translation [24,25,26,27]. Genetic analysis established that CTG18.1 expansion polymorphism has a stronger association with FECD compared to *TCF4* rs613872 polymorphism. In the unaffected population, fewer than 40 repeats were observed, while the globally recognized cut-off for a biological relevance with FECD is over 40 [11,22]. Expanded repeat length > 50 demonstrated a positive correlation with FECD, with 79% sensitivity and 96% specificity [28]. Hence, CTG18.1 repeat is a more specific index than the previously identified rs613872 polymorphism, whose specificity was 79%. Furthermore, CTG repeat length claims to have a potential prognostic value, as patients with a highly expanded CTC18.1 allele indicate a higher risk for keratoplasty and disease progression [29,30]. Herein, several studies have reached a consensus that the length of an abnormal CTG repeat could be the best predictor of disease [29]. As CTG repeats show ethnic diversity similar to SNPs, co-evaluation of rs613872 and CTG18.1 repeat length is promising for ameliorating the diagnostic value of genotyping in FECD [2,11,29,31].

The main purpose of the present research was to investigate the significance of *TCF4* CTG1.8 polymorphism as a risk factor for the occurrence of FECD in the Greek population. We provide practical advice for this cumbersome evaluation in order to avoid pitfalls and transform it into a cost-effective diagnostic algorithm. The need for standardization and use of reference materials for its accurate implementation is stressed. We also extend our previous work on the *TCF4* SNP [22] to a larger Greek FECD population and examine the impact of both intronic polymorphisms on the disease.

## 2. Results

The 36 patients (mean age 73 years, range 51–82 y, 58% female) and 58 healthy controls (mean age 74 years, range 42–87 y, 57% female) did not differ significantly when considering age and sex (the Mann–Whitney and x^2^ tests, both *p* > 0.05).

The evaluation was performed according to an established protocol according to the flow chart in Figure 1a. In addition, all the *TCF4* STR-PCR products of all samples were run in high-resolution agarose gel electrophoresis. Two distinct bands were observed in 16 FCED patients and 55 controls, one band in 17 FECD samples and 3 controls and no band in 3 FECD patients. Same products were then run in the fragment analysis mode of the DNA Sequencer, and better estimates of the sizes were obtained (in high accordance though with the sizes obtained empirically from the gel). All samples then underwent *TCF4* TP-PCR, which was followed by fragment analysis in the DNA Sequencer.

Regarding the samples with two bands in the gel, and therefore two peaks in the STR-PCR fragment analysis, there was high concordance with the estimated repeat sizes obtained from the two clusters of the TP-PCR analysis (due to non-normality, Spearman’s rho = 0.84), even in a polymorphic 14/17 sample differing by three triplets, or by only 9 bp, in the allelic products (Figure 2 and Figure 3). The reproducibility of STR size band estimation was very high (CV < 0.5%). In our dataset, all samples containing two distinct bands/peaks were assigned as normal N/N, as their STR-PCR products were less than 235 bp in size (corresponding to the 40 CTG triplets expected product length, which is the cut-off regarding abnormality as established in the literature, see Appendix A).

Regarding samples with only one band in the gel and one peak in STR-PCR fragment analysis, there were two types of results. In only one sample F24, the STR-PCR result was a single peak at 147 bp corresponding to 11 CTG triplets, and the TP-PCR fragment result was a single cluster, with the last peak at 87 bp corresponding again to 11 CTG triplets in the *TCF4* gene (Figure 4 and Figure 5, respectively). The sample was finally adjudicated as normal 11/11 homozygote by DNA Sequencing (Appendix A).

The other type of obtained result was the N/X, due to allelic dropout of the expanded X allele that did not produce a PCR product under the stringent conditions of the STR-PCR. In these cases, the TP-PCR method is the revealing method, with the appearance of a peak cluster protruding from a ladder-like fragment pattern (hence, the N number can be estimated). The same statement is valid for the three FECD cases, where no gel band was seen, and the result is denoted as expanded in both alleles (X/X). The expansion in Figure 6 goes up to 452 bp, which corresponds to 133 triplets in the *TCF4* gene (Appendix A); in these cases, it is difficult to estimate the exact number of the triplets in both chromosomes (the reproducibility in counting is 3–5%). DNA sequencing was performed in selected N/X and X/X cases, but the frameshift-like appearance in the electropherogram was very confusing in order to estimate accurately the number of repeats.

All samples provided *TCF4* genotyping results in both loci that were in accordance with the Hardy–Weinberg equilibrium (*p* > 0.05) (Table 1 and Table 2). Twenty out of 36 FECD patients (56%) possessed at least one expanded X allele (X allele frequency 32%) compared to 5% of healthy controls (3 out of 58, X allele frequency 3%). The odds ratio for developing the disease when carrying an expanded allele was estimated to be 19.85 (95% C.I. = 5.23–75.29) in the log-additive inheritance model. The frequency of *TCF4* risk G allele was increased to 40% in FECD patients compared to 17% in healthy subjects [odds ratio = 3.59 (95% C.I. = 1.68–7.69), log-additive]. The total number of samples is considered low as they provided intermediate statistical power (59%) to detect medium size effects (0.3) in χ^2^-tests as estimated by G*Power software; larger datasets are needed to corroborate findings. Haplotype analysis revealed strong linkage disequilibrium (D′ = 0.76, r = 0.52), with an OR of approximately 173 (but with no statistical confidence in the boundaries).

When examining both polymorphisms, a total of 19.5% of patients possessed a range of 3–4 pathological alleles in the two genetic *TCF4* loci, while 38.9% possessed 2 pathological alleles and could belong to a “grey zone” area.

## 3. Discussion

The main goal of our research was to designate the statistical significance of the association among *TCF4* CTG18.1 polymorphism and FECD prevalence in Greek population and—if possible—to suggest an algorithm that incorporates novel molecular modalities that guarantee a rational diagnostic pathway towards tailored therapy.

Since there is an extensive *TCF4* ethnic diversity reported in the literature, e.g., no SNP alterations or rarely in East Asian, South American and African populations [23], various percentages for CTG18.1 [27,32], our lab, which is the only one in Greece that investigates and performs FECD genetic analysis, opted to implement in-house molecular methods in order to detect and measure the prevalence of the *TCF4* CTG18.1 polymorphism (the SNP was investigated in our previous study [22]).

As suggested by other labs, both STR- and TP-PCR run in the fragment analysis mode in a genetic analyzer are recommended for an accurate CCG expansion evaluation. However, in our hands and in our dataset, it was observed that in about half of the cases (44.4%) when both alleles are normal (<40 repeats), a high-resolution agarose gel of the STR-PCR products could suffice in order to separate adequately the two bands. The samples are then assigned as N/N (when both bands are less than 235 bp), thus avoiding the need to use fluorescent fragment analysis and proceed to further cumbersome TP-PCR method. Indeed, in the normal samples, there is no need for the STR-PCR products to be run in fragment analysis, since in our study, we proved that STR-PCR gel optical estimation is similar with both STR- and TP- PCR fragment analysis (rho = 0.84).

In the rest of the cases, where one (or none) band is detected in the STR-PCR gel (mostly in patients, rarely in healthy subjects), there is a high suspicion for the presence of one (or two) abnormally expanded X alleles (due to allelic dropout). The lab should then proceed directly to TP-PCR fragment analysis. In the majority of these cases, a ladder-like appearance of the electropherogram is observed and experienced personnel can count the triplets and/or estimate the size of the normal allele in the case of a N/X sample. Thus, in this proposed algorithm seen in Figure 1b, the empirical and difficult expansion assay evaluation can become cost-effective. The yield outcome of this approach is the same as the established one (concordance 100%); only in three cases, there was a close proximity of the two bands in the gel that could be mistaken as one band and would merely result in an increased cost due to the addition of two fluorescent fragment analysis; no misclassification of the samples occurred. Larger datasets from us and others are needed to further validate this proposed algorithm in the future.

In the rare occasion where a single band is detected in the STR-PCR gel and a single cluster of peaks in the TP-PCR fragment analysis, there is suspicion for the presence of a homozygous sample that can be further verified by DNA Sequencing (optional).

In general, trinucleotide repeats are more difficult than tetra-, penta- or hexanucleotide in polymerase demands (no DNA slippage is allowed during polymerization) and in achieving accurate sizing. There is a need for standardization; no reference materials exist unlike, e.g., in Fragile X syndrome or Huntington’s disease, where commercial controls exist. Reference materials should include homozygotes, samples with two distinct normal alleles and X/X samples with an accurate estimate for the maximum number of triplets (target value ± standard deviation). In this sense, we identified one sample in our dataset that can be used as a reference material.

Twenty out of 36 FECD patients (56%) possessed at least one expanded X allele (X allele frequency 32%) compared to 5% of healthy controls (3 out of 58, X allele frequency 3%). Therefore, although diagnostic sensitivity of the X allele regarding FECD is low due to the multiparametric nature of the disease, diagnostic specificity reached 95%. The presence of the X allele was therefore verified in our Greek population in a significant proportion as in the rest of the European populations and can add value in the diagnostic and prognostic patient approach.

The odds ratio for developing the disease when carrying an expanded allele was estimated to be 19.85 (95% C.I. = 5.23–75.29) in the log-additive inheritance model. The frequency of *TCF4* risk G allele was increased to 40% in FECD patients compared to 17% in healthy subjects [odds ratio = 3.59 (95% C.I.= 1.68–7.69), log-additive]. Haplotype analysis revealed strong linkage disequilibrium (D′ = 0.76, r = 0.52), with an OR of approximately 173 (but with no statistical confidence in the boundaries).

When examining both polymorphisms, a total of 19.5% of patients possessed a range of 3–4 pathological alleles in the two genetic *TCF4* loci; therefore, in these cases, *TC4* gene expression is surely affected in both chromosomes.

Another 38.9% possess two pathological alleles and belong to a “grey zone” area where the alterations could matter depending on whether they are located at “cis” or “trans” position. It is expected that in half of these cases, the two pathological alleles are located in “trans”, with the severe possibility of altering and/or inactivating normal *TCF4* expression from both chromosomes.

However, in the rest 42% of the FECD Greek patients that possessed 0–1 pathological alleles, there is a strong possibility of mutations in other genes. As FECD is an oligogenic disease, the missing heritability could be attributed to other genes such as *ZEB1*, *AGBL1*, *SLC4A11* and *LOXHD1* that were detected in the initial studies. Genome-wide association studies have identified additional loci with *p*-values over 10^−7^ besides *TCF4*: *KANK4* (KN motif- and ankyrin repeat domain-containing protein 4) gene, *LAMC1* (laminin gamma 1) gene and *ATP1B1* gene (Na^+^/K^+^ transporting ATPase, beta-1 polypeptide) [31]. Also, other genes are under investigation such as *CLU*, *FAS* and DNA repair genes (*RAD51*, *XRCC1*, *FEN1*, *NEIL1*), etc. [12,27,33].

With the advent of next-generation sequencing techniques, which are becoming continuously more economic and efficient, a targeted panel with all the aforementioned genes or a whole exome approach could result in the full genetic elucidation for FECD. This could lead to personalized precision medicine therapeutics such as gene editing, anti-sense and mRNA technologies in order to correct the genetic defect or alter the expression of an important FECD gene [27].

## 4. Materials and Methods

### 4.1. Patients

Our study was conducted on a cohort of 36 well-ascertained Greek patients with late-onset FECD (Krachmer scale > 2) and 58 healthy controls, who were investigated during a two-year period in “G. Gennimatas” General Hospital. All participants were recruited after ophthalmological evaluation and clinical data collection. Clinical evaluation was based on visual acuity and slit-lamp anterior segment examination. A modified version of the Krachmer scale classification system was used to determine disease severity, grading patients on a scale of 0 to 5. Grade 0: no central cornea guttae, grade 1: scattered central cornea guttae, grade 2: 1 or 2 mm of central cornea guttae, grade 3: 2 to 5 mm of grouped cornea guttae, grade 4: 5 mm of grouped central cornea guttae and grade 5: cornea guttae with corneal edema [22]. Participants were selected as patients only when the grade was 2 or more. All individuals agreed to sign an informed consent before blood sampling. Peripheral blood samples were collected after approvals from the “G.Gennimatas” General Hospital Scientific and Bioethics committees.

### 4.2. Genomic DNA Isolation

Peripheral blood samples were collected in EDTA tubes, and after centrifugation, the buffy coat (white cells) was stored at −20 °C until DNA extraction. DNA isolation was performed using the High Pure PCR Template Extraction Kit (Roche Applied Science, Basel, Switzerland) according to the manufacturer’s instructions. DNA concentration was calculated by the Quant-IT dsDNA BR Assay Kit in a Qubit fluorimeter (ThermoFisher Invitrogen, Waltham, MA, USA), which employs a dye specific for DNA. DNA was stored at −20 °C until further use.

### 4.3. TCF4 CTG Expansion Evaluation by STR-PCR and TP-PCR Methods

All PCR reactions were run in the Eppendorf Mastercycler gradient cycler with 2X GoTaq G2 Green Master mix polymerase (Promega, Madison, WI, USA). Primers and PCR conditions for the two expansion evaluation methods were as described in Mootha et al. [34].

Primers P1 and P2 were used for Short Tandem Repeat PCR (STR-PCR). PCR products were run in a high-resolution 4% agarose (1:1 with Nusieve) and, in parallel, in the fragment analysis mode of a SeqStudio genetic analyzer (in the latter case, the P1 primer is labeled with fluorescein). The expected size of the STR-PCR allelic products—depending on the number of CTG triplets they contain—is provided in Appendix A. The size is estimated with the use of appropriate MW (molecular weight) controls for each of the two running modes. The GenBank sequence for the *TCF4* gene (NG_011716.2) contains 24 CTG repeats (or 25 TGC repeats; hence, the other name that some authors use in order to define this polymorphism [24,25,26].

Fluorescent P1 along primers P3 and P4 are used for the triplet-repeat primed PCR (TP-PCR) [35]. As depicted in Figure 7, primer P4 carries in its 3′ end a complementary sequence to five CTG triplets. This structure results in the possibility of many annealing sites—starting with the first CTG triplet in *TCF4* sequence in each chromosome—and creates many PCR products differing in size by multiples of 3 bp. The expected size of the TP-PCR allelic products, depending on the number of CTG triplets they contain, is provided in Appendix A. TP-PCR is followed by fragment analysis in the genetic analyzer SeqStudio (Applied Biosystems, ThermoFischer Scientific, Waltham, MA, USA), and triplets are seen as different peaks and are counted with the Peak Scanner software v.2 (https://www.thermofisher.com/order/catalog/product/4381867, accessed on 19 November 2025). In theory, the triplets should ideally result in peaks of equal intensity from each chromosome (Figure 1); however, diminishing intensities are usually seen with a ladder-like appearance. In general, pathological expansion, noted by X symbol, is defined when at least one of the two alleles, after accurate triplet counting, contains > 40 triplets.

### 4.4. DNA Sequencing

In the case of a sample where a single band is seen on gel electrophoresis and then a single cluster in the TP-PCR fragment analysis, cycle sequencing of the corresponding purified STR-PCR product is performed with primer P1 and BigDye Terminator vs3.1 and run in an ABI SeqStudio genetic analyzer in order to confirm homozygosity.

### 4.5. TCF4 rs 613872 Real-Time-qPCR Assay

Our previously developed real-time qPCR-melting curve method in the LightCycler platform (Roche, Basel, Switzerland) [22] was used for the genotyping of the rs613872 *TCF4* polymorphism (g.97923 A > C, T > G in the other DNA strand that is being investigated by the probes of this method) in 14 additional FECD patients (thus, for a total of 36 patients).

### 4.6. Statistics

Statistical analysis was performed using SPSS 26.0 software package for Windows (IBM). The criterion used for statistical significance was *p* < 0.05. Normality of distribution of age was assessed with the Kolmogorov–Smirnov test. Median age between patients and controls was compared with the Mann–Whitney test and sex proportions with the chi-squared test. Correlation of peaks in fragment analysis was performed with Spearman’s test. The online SNPstats software tool (https://www.snpstats.net/, accessed on 19 November 2025) [36] was used in order to examine conformance with Hardy–Weinberg equilibrium (HWE), allelic count, genotypic frequencies, haplotype analysis and odds ratio (OR) calculation with 95% confidence intervals (C.I.) for all plausible inheritance models (dominant, co-dominant, recessive or log-additive). The significant model with the lower Akaike information criterion (AIC) was selected. Adequacy of the number of total samples and statistical power for all χ^2^-tests was estimated with the G*Power 3.1.9.6 software (written by Franz Faul at the University of Kiel, Kiel, Germany).

## 5. Conclusions

This study confirms the strong association of both the *TCF4* CTG18.1 trinucleotide repeat expansion and the rs613872 SNP with late-onset FECD in the Greek population. The proposed dual-step STR-PCR/TP-PCR workflow provides a validated, cost-effective molecular algorithm for accurate genetic diagnosis. The significant odds ratios observed highlight the major role of *TCF4* variants in FECD susceptibility, whilst incomplete genetic penetrance observed suggests that additional genes contribute to disease heritability, warranting further genomic investigation. Although this work supplements molecular diagnostics for FECD towards the implementation of precision-medicine approaches in corneal dystrophies, our findings need international standardization of *TCF4* expansion testing and the establishment of reliable reference materials.

## Figures and Tables

**Figure 1 ijms-26-11356-f001:**
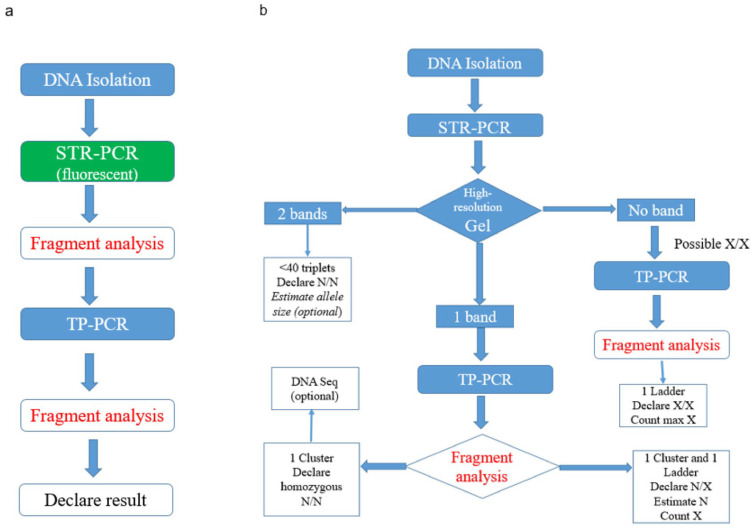
(**a**) Existing method flow chart for the evaluation of expansion in the *TCF4* gene. (**b**) Proposed cost-effective diagnostic algorithm for the same evaluation.

**Figure 2 ijms-26-11356-f002:**
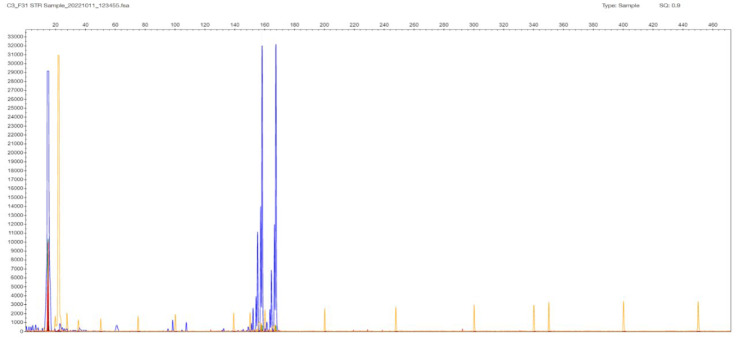
F31 patient sample STR-PCR result with 2 peaks at 159 and 167 bp corresponding to 14 and 17 CTG triplets in the *TCF4* gene. The sample is characterized as 14/17, and it is normal in terms of expansion in both chromosomes (N, N). Liz GS 500 (ABI) is used as a MW size standard in all fragment results (depicted as low-intensity orange vertical lines spanning the whole electropherogram).

**Figure 3 ijms-26-11356-f003:**
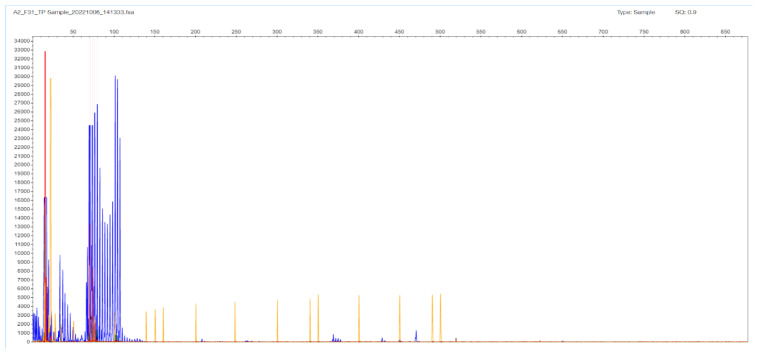
F31 patient sample TP-PCR results with two clusters of bands with peaks at 90 and 102 bp corresponding to 12 and 17 CTG triplets in the *TCF4* gene.

**Figure 4 ijms-26-11356-f004:**
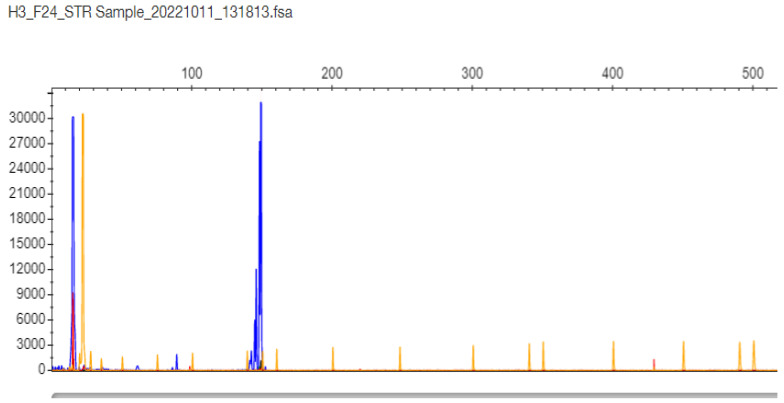
F24 patient sample STR-PCR results with a single peak at 147 bp corresponding to 11 CTG triplets in the *TCF4* gene (the sample is proven as normal 11/11 homozygote by DNA Sequencing).

**Figure 5 ijms-26-11356-f005:**
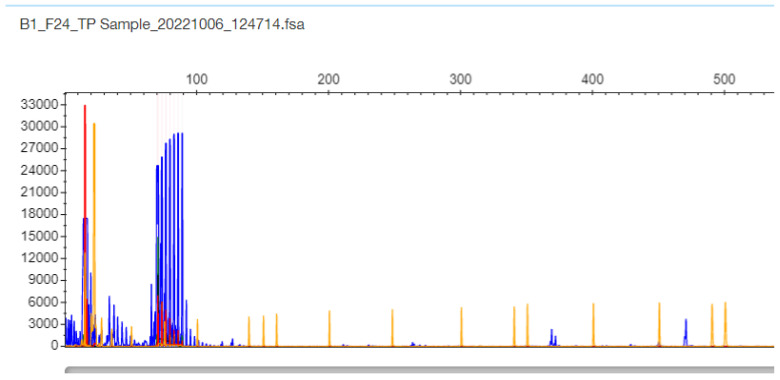
F24 patient sample TP-PCR results with a single cluster with the last peak at 87 bp corresponding to 11 CTG triplets in the *TCF4* gene (the sample is proven as normal 11/11 homozygote by DNA Sequencing).

**Figure 6 ijms-26-11356-f006:**
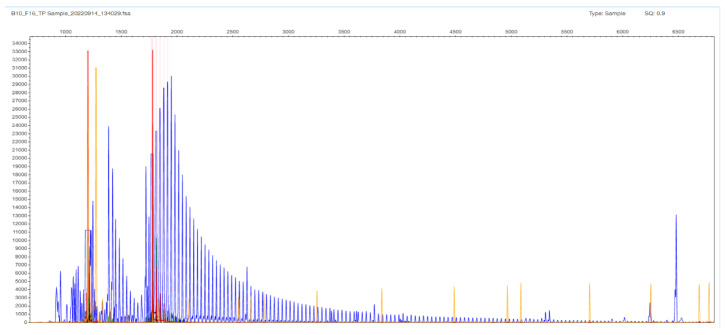
F16 patient sample with no gel band or peak in the STR-PCR but with a ladder appearance in the TP-PCR fragment analysis image. The expansion goes up to 452 bp, which corresponds to 133 triplets in the *TCF4* gene. The result is denoted as expanded (X/X); it is difficult to estimate the exact number of the triplets in both chromosomes.

**Figure 7 ijms-26-11356-f007:**
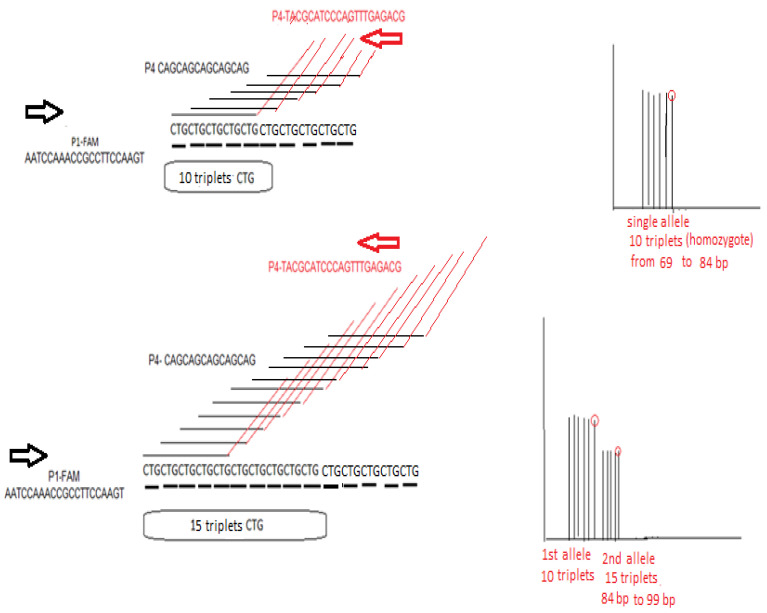
An explanatory figure of TP-PCR expansion assay in the case of *TCF4* trinucleotide CTG repeat estimation. P1-FAM is the forward primer, and P4 is the reverse primer that contains a “black” part which is 100% complementary to five CTG triplets in a row and a “red” part that is irrelevant and same in sequence as the P3 primer (sequences are provided 5′ → 3′). In the left part of the figure, the stochastic action of P4 primer in initiating multiple PCR start sites is depicted in two cases: in the upper part, in a chromosome containing 10 CTG repeats, where there is equal possibility of creating six PCR products of equal intensity between 69 and 84 bp seen as peaks in the fragment analysis (differing by 3 bp), and in the lower part in a chromosome containing 15 CTG repeats, where there is equal possibility of creating eleven PCR products of equal intensity. In the upper right part of the figure, the expected fragment analysis result is seen for a homozygous 10/10 sample. In the lower right part of the figure, an ideal expected fragment analysis result is seen for a heterozygous 15/10 sample, where there will be two peak clusters of different intensity: the first cluster contains six peaks corresponding in the 10 triplets being present in both chromosomes and another cluster of peaks of weaker half intensity, ranging from 87 to 99 bp, corresponding to the rest of the triplets in only one of the two chromosomes.

**Table 1 ijms-26-11356-t001:** *TCF4* expansion assay results analyzed with SNPStats software for odds ratio adjusted by sex and age (N denotes normal triplet result, X abnormal triplet result > 40, STATUS = 0 healthy controls, STATUS = 1 FECD patients, NA non-applicable).

Model	Genotype	STATUS = 0	STATUS = 1	OR (95% CI)	*p*-Value	AIC
Codominant	N/N	55 (94.8%)	16 (44.4%)	1.00	<0.0001	103
	N/X	3 (5.2%)	17 (47.2%)	19.41 (5.02–75.02)		
	X/X	0 (0%)	3 (8.3%)	NA (0.00–NA)		
Dominant	N/N	55 (94.8%)	16 (44.4%)	1.00	<0.0001	102
	N/X-X/X	3 (5.2%)	20 (55.6%)	22.83 (5.99–87.09)		
Recessive	N/N-N/X	58 (100%)	33 (91.7%)	1.00	0.015	127
	X/X	0 (0%)	3 (8.3%)	NA (0.00–NA)		
Log-additive	---	---	---	19.85 (5.23–75.29)	<0.0001	101

**Table 2 ijms-26-11356-t002:** *TCF4* rs613872 SNP results analyzed with SNPStats software (STATUS = 0 healthy controls, STATUS = 1 FECD patients) for odds ratio (adjusted by sex and age). G is the risk allele.

Model	Genotype	STATUS = 0	STATUS = 1	OR (95% CI)	*p*-Value	AIC
Codominant	T/T	38 (65.5%)	14 (38.9%)	1.00	2 × 10^−4^	118
	G/T	20 (34.5%)	15 (41.7%)	2.11 (0.81–5.45)		
	G/G	0 (0%)	7 (19.4%)	NA (0.00–NA)		
Dominant	T/T	38 (65.5%)	14 (38.9%)	1.00	0.0082	126
	G/T-G/G	20 (34.5%)	22 (61.1%)	3.26 (1.33–8.02)		
Recessive	T/T-G/T	58 (100%)	29 (80.6%)	1.00	1 × 10^−4^	118
	G/G	0 (0%)	7 (19.4%)	NA (0.00–NA)		
Log-additive	---	---	---	3.59 (1.68–7.69)	4 × 10^−4^	120

## Data Availability

The original contributions presented in this study are included in the article/Appendix A. Further inquiries can be directed to the corresponding authors.

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
