# Peer review of "Molecular Studies of TCF4 Gene and Correlation with Late-Onset Fuchs Endothelial Corneal Dystrophy in the Greek Population: A Novel Cost-Effective Diagnostic Algorithm"

_ijms, 2025, doi:10.3390/ijms262311356_

Round 1
Reviewer 1 Report
Comments and Suggestions for Authors
Petri and colleagues present a study of TCF4 gene mutations associated with Fuchs endothelial corneal dystrophy (FECD). The authors compared DNA samples of white blood cells from late onset FECD patients and healthy donors using STR-PCR and triplet-repeat primed PCR followed by gel electrophoresis. They found significantly more frequent presence of an expanded allele of TCF4 gene with more than 40 trinucleotide repeats among the group of FECD patients. These results may be used for the prediction of FECD development. This is a solid study with potential medical implementations.
Author Response
Thank you for your remarks.
Reviewer 2 Report
Comments and Suggestions for Authors
This manuscript presents an investigation into the association between TCF4 gene polymorphisms (CTG18.1 repeat expansion and rs613872 SNP) and late-onset Fuchs endothelial corneal dystrophy (FECD) in a Greek population. While the topic is relevant and potentially valuable to molecular ophthalmology, the manuscript in its current form suffers from serious conceptual, methodological, and structural issues that must be addressed before it can be considered for publication.
- The cohort is far too small to yield meaningful statistical inference, especially given the variability in repeat expansion lengths. Confidence intervals are excessively wide, and no power calculation is presented. Without robust justification for the sample size, the conclusions appear overstated.
- The design of the picture is unreasonable, including the font of the coordinates in the picture and the clarity of the picture.
- The method introduction of this manuscript is too simple, especially the lack of a clear schematic diagram of the method principle.
- The data structure is simple and lacks cross-validation data.
- The so-called “cost-effective diagnostic algorithm” is insufficiently defined and lacks validation. The proposed flowchart is speculative and not experimentally tested. The authors must provide quantitative evidence (e.g., sensitivity, specificity, reproducibility) demonstrating that the algorithm indeed improves diagnostic efficiency.
Round 2
Reviewer 2 Report
Comments and Suggestions for Authors
The authors have responded to my concerns